# *Artemisia arborescens* (Vaill.) L.: Micromorphology, Essential Oil Composition, and Its Potential as an Alternative Biocontrol Product

**DOI:** 10.3390/plants13060825

**Published:** 2024-03-13

**Authors:** Flavio Polito, Marina Papaianni, Sheridan Lois Woo, Paola Malaspina, Laura Cornara, Vincenzo De Feo

**Affiliations:** 1Department of Pharmacy, University of Salerno, Via Giovanni Paolo II 132, 84084 Fisciano, Italy; fpolito@unisa.it (F.P.); defeo@unisa.it (V.D.F.); 2Department of Agricultural Sciences, University of Naples Federico II, Via Università 133, 80055 Portici, Italy; marina.papaianni@unina.it; 3Department of Pharmacy, University of Naples Federico II, Via Domenico Montesano 49, 80131 Naples, Italy; woo@unina.it; 4Department of Earth, Environment and Life Sciences, University of Genova, Corso Europa 26, 16132 Genova, Italy; paola.malaspina@unige.it

**Keywords:** Asteraceae, glandular trichomes, secretory ducts, phytochemistry, phytotoxicity, antimicrobial activity

## Abstract

*Artemisia arborescens* is a Mediterranean evergreen shrub, with silver grey-green tomentose leaves and a strong scent. It has various ethnopharmacological uses and its secondary metabolites have demonstrated antimicrobial, antiviral, pharmaceutical, phytotoxic, and insecticidal activities. Different extracts obtained from aerial parts of this species are known for their allelopathic effect, but similar studies on its essential oil (EO) are lacking. Therefore, we carried out a pharmacognostic study, obtaining the characterization of the secretory structures and the EO produced. Trans-thujone and camphor are the main components, followed by aromadendrene, camphene, and 8-cedren-13-ol. EO phytotoxic activity was tested on weed plants (*Lolium multiflorum* Lam. and *Sinapis arvensis* L.) and crops (*Raphanus sativus* L. and *Cucumis sativus* L.), showing inhibition on both germination and radical growth of the two weeds tested. The effects of the EO against the bacterial plant pathogens *Xanthomonas campestris* pv. *campestris* (Gram−) and *Pseudomonas syringae* pv. *tomato* (Gram+) was also assayed. The minimum inhibitory concentration (MIC) was observed when it was used undiluted [100% *v*/*v*], and growth inhibition when diluted at different doses. The antimicrobial activity was also confirmed by the cellular material release and biofilm formation assays. The overall data show that *A*. *arborescens* EO can find application as a potential alternative biocontrol product against weeds and plant pathogens. This goal is particularly important from the perspective of replacing synthetic pesticides with natural products, which safeguard both the environment and the health of consumers.

## 1. Introduction

The genus *Artemisia* (Asteraceae) includes about 500 species mainly distributed in the temperate zones of the Northern hemisphere [1], most of which have been traditionally used in medicine since ancient times for their broad therapeutic potential [2].

*Artemisia arborescens* (Vaill.) L., also known as silver sage and tree wormwood, is an endemism of the Mediterranean area. This multi-branched shrub has silver grey-green, deeply divided leaves, covered by a dense indumentum, and is characterized by a strong scent. It is mainly used in the culinary and alcoholic beverages industries [3], but it has been also employed for various ethnopharmacological purposes, such as in traditional medicine, e.g., for the treatment of digestive and respiratory problems, as a febrifuge for domestic uses against moths and bed bugs, and as a magical/ritual plant [4,5,6].

The plant extract contains several secondary metabolites, which have demonstrated different biological activities such as an effect on rat-isolated smooth muscle [7] and feeding deterrent properties [8]. The essential oil (EO) obtained from plants of *A. arborescens* growing in Sicily with β-thujone and the sesquiterpene hydrocarbon chamazulene as the main compounds, had inhibition ability against *Lysteria monocytogenes* strains [9]. This essential oil also presented anti-herpes virus activity, being able to inactivate the virus and inhibit cell-to-cell virus diffusion [10].

Moreover, some *Artemisia* species have been reported for their allelopathic potential [11,12,13]. Regarding *A. arborescens*, different extracts obtained from aerial parts [14,15,16] and from the maceration of leaf litter [17] have shown allelopathic effects. However, the biological properties of *A. arborescens* EO have been little explored [18]. In particular, there is a lack of data on its phytotoxic effect on weeds, as well as on its antimicrobial activities against pathogenic bacteria that affect agricultural crops. Crop protection is currently in a transitional phase, in which synthetic chemicals are still widely used, but there is a need for their reduction in farm management [19]. Therefore, new practices must be gradually introduced in order to take into account both environmental safety and socio-economic factors [20]. In recent years, the growth in productivity and international trade has also led to a growing incidence of some diseases, causing an increase in the use of pesticides, which are a risk to the environment and agricultural ecosystems [21].

There are various methods to control pathogens, but some of them pose risks for developing resistance in the pathogen population. For instance, selection of specific genetic pathways in plant breeding to improve disease resistance are practical but may lead to the emergence of virulence genes in the pathogens [22]. On the other hand, biological control involving microorganisms can be an effective way to reduce the negative impact of synthetic chemicals on the environment and minimize pollution [23,24]. To promote sustainable agriculture, several countries have adopted a protection plan to reduce the use of pesticides by up to 50%; however, there are still few valid alternatives [25]. Therefore, it is important to increase our understanding of the different mechanisms involved in biocontrol to improve its effectiveness and extend its use.

Several studies have demonstrated that biopesticides derived from essential oils can be used as selective herbicides [26,27] and for their antibacterial and antifungal properties, making them ideal for the protection of plants and crops [28].

Therefore, the aim of this study was to carry out a pharmacognostic characterization of leaves and branches of *A. arborescens*, to obtain the phytochemical profile of the EO. Afterwards, the phytotoxicity of the EO was tested on both weed and crop plants, while its antimicrobial activity was assayed against two bacterial plant pathogens in order to evaluate its potential use as a new biocontrol product in agriculture.

## 2. Results

### 2.1. Anatomical and Micromorphological Analyses

*A. arborescens* is a branched shrub (Figure 1a) with petiolate deeply divided leaves (Figure 1b), showing a dense cover of trichomes that gives them a silver grey-green colour (Figure 1c). Details of the anatomical and micromorphological features of the leaf blade, petiole, and young stem (small branches) were highlighted by both light microscopy (LM) and scanning electron microscopy (SEM) analyses (Figure 2, Figure 3 and Figure 4).

#### 2.1.1. Leaves

On both surfaces, many trichomes are visible (Figure 2a; Figure 3a), mainly consisting of T-shaped non-glandular trichomes (NGTs) (Figure 2b; Figure 3b). These trichomes show a neck made of a variable number of cells bearing a terminal cell with long straight arms (Figure 2b,c; Figure 3a–c). In NGTs, TBO staining highlights, in greenish-blue, the presence of suberin- or cutin-like hydrophobic substances on the side walls of the neck cells (Figure 2c, black arrow). Capitate glandular trichomes (GTs) also occur on both surfaces of the leaf blade, deeply embedded in the epidermis and partially hidden by the dense covering of the NGTs (Figure 2c,d, red arrows; Figure 3a,c, red arrows). GTs are also difficult to individuate in transversal sections clarified with Chloral Hydrate (Figure 2d, red arrow). They can be better seen on the leaf surface after bleaching with a solution of sodium hypochlorite since the yellow-brown EO is maintained (Figure 2e). Glandular trichomes show a sac-like structure, with a short neck bearing a big glandular head covered with a thin cuticle sheath (Figure 2c,d and Figure 3d,e) and are of biseriate type (Figure 2f and Figure 3f). The secretion within the glandular head positively reacts with Fluoral Yellow (FY), revealing the presence of lipophilic substances. In the leaf transversal sections embedded in paraffin, stained both with TBO (Figure 2h, red arrow) and Haematoxylin–Eosin (Figure 2i, red arrow), the secretory ducts are well visible and appear located near the xylematic portion of the vascular bundles.

#### 2.1.2. Petiole and Young Stem

The leaf petiole shows a trapezoidal shape and a dense indumentum (Figure 4a). The mono-stratified epidermis is covered by a thick cuticle stained in red by Sudan III (Figure 4b, red arrow) and in bright yellow by FY (Figure 4c, red arrow), revealing the presence of lipophilic substances. Under the epidermis, collenchyma and chlorenchyma are alternately arranged, and one central and two lateral vascular bundles are well visible (Figure 4a,b). Two secretory ducts are present on the sides of the main vascular bundle, in the central zone of the petiole (Figure 4b black arrows). The secretion inside them reacts positively with FY, appearing bright yellow (Figure 4c, white arrows). In addition, in the transversal section of the young stem, many large secretory ducts can be found (Figure 4d, white arrows) located in the cortex, near the vascular bundles and arranged in a circle. Also, in this case, the secretion within the secretory ducts is stained in bright yellow by FY (Figure 4e, white arrows). Below the epidermis, the older stem is characterized by a continuous periderm consisting of two to three layers of enlarged cells arranged in radial rows (Figure 4f, arrow). SEM analysis at high magnification permitted the individuation of the presence of GTs also on the young stem surface (Figure 4g, white arrows; the insert shows, with a white circle, the zone where the GTs are individuated). Moreover, inside the parenchyma cells of the pith, the presence of small crystal druses is detected (Figure 4h, white arrows).

### 2.2. Chemical Composition of EO

The EO yield was 0.12% on a dry weight basis, in agreement with the EO yield of other species of *Artemisa* [29].

The analysis of the EO allowed us to identify a total of 51 components, corresponding to 96.3% of the total EO. Monoterpenes (25 components) predominated in the EO (67.0%), and oxygenated monoterpenes were the most representative class (49.0%). Among oxygenated monoterpenes *trans*-thujone (24.2%) and camphor (18.9%) were the main components of the EO. Among monoterpene hydrocarbons, the main components were camphene (6.0%), *p*-cymene (2.4%), sabinene (2.4%), and α-pinene (2.3%). The sesquiterpene fraction was almost equally divided among hydrocarbons (14.9%) and oxygenated compounds (13.9%). Aromadendrene (6.5%) was the main component among sesquiterpene hydrocarbons, *trans*-nuciferol (4.2%) and caryophyllene oxide (3.0%) were the main components among oxygenated sesquiterpenes (Table 1).

### 2.3. Phytotoxic Activity

Table 2 shows the activity of the EO on germination and radical growth of the tested plant species. The EO demonstrated variable phytotoxicity depending on the species and the concentration used. It was effective in counteracting the radical growth of weeds but less effective in preventing their germination. In fact, at 1000 µg/mL, the germination of *L. multiflorum* was completely inhibited, whereas *S. arvensis* was not affected. At 1000 and 500 µg/mL, the radical elongation of *L. multiflorum* and *S. arvensis* was completely inhibited, while at lower concentrations, the root inhibition was minor. Other concentrations exerted a weaker effect. Regarding food crops, *R. sativus* was found to be very sensitive to the EO, which exerted a complete inhibition of germination at concentration of 1000 and 500 µg/mL and complete inhibition of radical growth also at 250 µg/mL. *C. sativus*, on the other hand, was much less sensitive to the EO for radical growth and especially for germination.

### 2.4. Antimicrobial and Antibiofilm Activity of the Essential Oil

All EO dilutions tested were able to lyse both *Xcc* and *P. syringae* plant pathogens. On soft agar with the bacteria, the EO spotting consistently formed clearing zones of approximately 1 cm in diameter (Figure 5) showing a halo of inhibition when the EO was used pure (100% *v*/*v*) and at 10% *v*/*v* dilution. In the antimicrobial assay, the EO was tested for each bacterial strain at different dilutions, demonstrating a MIC value when used undiluted and 90%, 80%, and 76% for the serial dilutions (Figure 6). The EO showed the same antimicrobial activity in a dose-dependent manner for both the Gram+ (*P. syringae*) and Gram− (*Xcc*) bacteria.

In addition to the antimicrobial activity, the activity of the EO on biofilm formation was performed on *Xcc*, a biofilm-producing bacterium. The results shown in Figure 7 demonstrated that all the dilutions tested have the same ability to reduce the biofilm biomass by about 80%.

### 2.5. Activity of EO in Planta

Tomato (*Solanum lycopersicum*) plantlets were treated by spraying the aerial vegetative parts with suspensions of the EO at different concentrations and a commercial pesticide. After three days, the plants were infected by spraying with *P. syringae*. The results of the trial showed a statistically significant decrease in disease symptoms (84%) with treatment at 10% *v*/*v* of the EO, but the effect was slightly phytotoxic to the plants. Interestingly, when the plants were treated with 0.1% *v*/*v* of the EO, the decrease in disease symptoms on the plants was comparable to that of the commercial pesticide treatment (65%) (Figure 8), without phytotoxic symptoms. As expected, the plants from the H_2_O control did not display any symptoms of the disease.

## 3. Discussion

In the Asteraceae, besides having secretory ducts inside the organs, another secreting system is represented by glandular trichomes on the organ surfaces [30]. In *A. arborescens*, the aerial parts show a dense cover of trichomes, mainly represented by non-glandular T-shaped trichomes (NTGs), which helps the plant to adapt to dry conditions typical of Mediterranean coasts. These structures play a role in reflecting high UV and visible radiation as well as in the control of water loss and temperature regulation [31]. Concerning secretory tissues, *A. arborescens* shows both secretory ducts and glandular trichomes (GTs). These secretory structures are involved in essential oil production and have a significant role in the plant responses to biotic and abiotic stresses [32], as well as in plant defence against herbivores and pathogens [33,34]. Many studies documented that the secretion of both secretory ducts and trichomes consists of several compounds, such as essential oils, lipids, sesquiterpene lactones, resins, pectin-like substances, alkaloids, flavonoids, and tannins [35,36,37,38,39].

On the leaf and young stem surface, the presence of a dense cover of NTGs obstructs the observation of GTs, similar to that reported for other *Artemisa* species, such as *A. nova* Nelson [40] and *A. umbelliformis* Lam. [41], and for other Asteraceae, i.e., *Santolina impressa* Hoffmanns. & Link [42]. In addition, the location of these trichomes in depressions in the leaf epidermis made it difficult to detect the biseriate structure, a typical characteristic of the glandular trichomes of many representatives of the Asteraceae and referred for different species of *Artemisia* [35,40,43,44].

Our observations confirm the presence of very large secretory ducts in the stem cortex, while only small ducts are present in the leaf base parenchyma and in the petiole parenchyma, according to data reported by Janaćković et al. [44]. These authors for the first time described the secretory structures of *A. arborescens*, comparing them with those of other related species. In agreement with the observations made by these authors for both *A. arborescens* and *A. campestris*, we also found in the stem the presence of a well-developed periderm [44]. Regarding the presence of small crystals inside the pith parenchyma cells, our analyses by SEM at high magnification showed very small crystals druses instead of rhomboidal crystals previously referred to by Janaćković et al. [44]

Some studies are available in the literature regarding the composition of the EO of *A. arborescens*, and some of these concern the EOs obtained from species growing in Italy. A study by Ornano et al. [45] analysed an EO from *A. arborescens* growing in Sardinia and collected at different times of the year. In this EO, the most present classes were oxygenated monoterpenes (37.7–57.0%) and sesquiterpene hydrocarbons (32.0–55.3%). The main components were *trans*-thujone, with amounts ranging from 33.8 to 53.2%, chamazulene (25.6 to 51.5%) and germacrene D (3.2–5.4%). This composition agrees only in part with data reported in our work: *trans*-thujone was the main component also in this case but in lower amounts (24.2%) and germacrene D was present for only 1.5%. Chamazulene, however, was completely absent. The absence of chamazulene could be due to the genetic background and environmental conditions [46] and to the fact that the plant was harvested far from its flowering time, which is the most appropriate vegetative stage to obtain the maximum amount of chamazulene [6]. Militello and others [9] studied an EO from a Sicilian sample of *A. arborescens* that was rich in oxygenated monoterpenes (57.3%) and sesquiterpene hydrocarbons (27.1%). The main components were *trans*-thujone (45.0%), chamazulene (22.7%), camphor (6.8%) and germacrene D (3.3%). This composition also partially agrees with data reported in the present work, as *trans*-thujone is the main component but in lower quantities (24.2%), chamazulene is absent, camphor is present but as the second main component (18.9%), and germacrene D was present in lower amounts (1.5%) Finally, Presti and coworkers [47] reported the composition of three EOs of *A. arborescens* from Calabria, Sicily, and the island of Lipari. These EOs were rich in camphor (20.1–39.5%) and chamazulene (27.1–37.6%). Between these two components, only camphor is also present in the EO studied in this work but in lower quantities (18.9%), while chamazulene is absent. In these EOs, *trans*-thujone was present only in the EO coming from Calabria (1.5%) and from Lipari Island (6.6%), but in much lower amounts than that found in our work (24.2%), where it represented the main component.

Very few studies in the literature focused on the allelopathic and herbicidal activity of *A. arborescens* EO. Dudai and co-workers [18] reported the EO activity of *A. arborescens* on wheat seeds, highlighting a reasonable ability to inhibit germination. However, some extracts of the plant have been reported for their phytotoxicity on *Lactuca sativa* L. [14]. In addition, a methanolic extract obtained from the maceration of leaf litter released by the plant showed remarkable phytotoxicity [17]. The EO tested in our work confirms the presence of allelopathic activity mentioned in the literature, although with different mechanisms based on the species considered. On weeds such as *L. multiflorum*, the tested EO acted on both germination and root growth, while on *S. arvensis*, it was mainly active in counteracting root growth. The activity on species of food interest was also variable as the EO used in this work has allelopathic activity on *R. sativus* both for germination and root growth but is not very active on either the germination or root growth of *C. sativus*.

The different biological responses observed can be explained by the phenomenon called hormesis, a two-phased biological response in which a low dose of a biological agent shows inhibitory activity and a high dose shows the contrary effect [48]. Hormesis has widely been reported in allelopathic and phytotoxic activity of plant secondary metabolites [49].

To find and ascertain the antibacterial activity of essential oils, recently, numerous studies have been carried out [50]. Although the exact mechanism of action is still unknown, several investigations have indicated that essential oil components may enter cells and disrupt cellular metabolism [51]. Phenols, such as eugenol and carvacrol, damage cellular membranes and interact with enzyme-active sites. The lipid bilayer of bacteria may absorb EOs and their constituents, that interact with the cell membrane, filling in the gaps between the chains of fatty acids [52]. Several EOs from the genus *Artemisia*, as well as their main components, have shown antibacterial properties [53]. Additionally, it has been noted that the most prevalent classes of components in the essential oil of *A*. *arborescens*, including oxygenated monoterpenes and hydrocarbon sesquiterpenes, have antibacterial properties [54,55]. Moreover, the volatile phase of essential oils of different plants was also reported to possess more antimicrobial activity against plant pathogenic bacteria [27,56]. Some investigators reported that the antimicrobial activity resulted from a direct effect of essential oil vapours on the bacteria. In our case, it could be hypothesized that *A*. *arborescens* EO presented a better effect as a protective treatment because it was applied before pathogen inoculation. To the best of our knowledge, this study is the first to document the antibacterial properties of *A. arborescens* EO in vitro and *in planta*. The EO demonstrated antimicrobial activity in vitro in a dose-dependent manner on both the Gram+ (*P. syringae*) and Gram− (*Xcc*) bacteria and caused a reduction of 80% in the biofilm formation by *Xanthomonas campestris* pv. *campestris*. *In planta*, the treatment with 0.1% *v*/*v* of the EO was the most promising result since a decrease in disease symptoms on tomato plants was comparable to that obtained with the commercial pesticide. Furthermore, at this concentration of the EO, no symptoms of phytotoxicity were observed in the plants. Overall, our data indicated that the EO of *A. arborescens* has variable phytotoxicity on weeds or crops and can inhibit the growth of plant pathogenic bacteria responsible for significant crop losses. Nowadays, the search for new natural products that can replace synthetic pesticides to safeguard the environment and the health of consumers is becoming increasingly popular. Recent studies highlight the important role that EOs may play as biopesticides both used as selective herbicides to control weeds and against bacterial or fungal pathogens, making them excellent candidates for the protection of plants and crops [26,27,28]. Therefore, the findings collected in our study opens interesting perspectives for the use of *A. arborescens* EO in organic agriculture, paving the way for the development of new phytosanitary treatments.

## 4. Materials and Methods

### 4.1. Plant Materials

Samples of leafy young stems of *Artemisia arborescens* (Vaill.) L. were collected at Hanbury Botanical Gardens of Ventimiglia (IM; Italy) during March 2023, before the flowering stage. The species was identified by Prof. L. Cornara. A voucher specimen (GDOR60969) was deposited at the herbarium of the Natural History Museum Giacomo Doria of Genova (Italy).

### 4.2. Light and Scanning Electron Microscopy

For the light microscopy (LM), small leaf samples were bleached with a commercial solution of sodium hypochlorite 2.2% to detect glandular trichomes containing the EO. Handmade cross-sections of fresh leaves and stems were made by using a double-edged razor blade and then cleared with an aqueous solution of chloral hydrate and mounted in a chloral hydrate–glycerol solution to prevent crystallization of the reagent, following Jackson and Snowdon [57]. This technique allowed us to better characterize leaf and stem anatomical structures and tissues. Fresh sections of leaves, petioles, and young stems were also stained with both Sudan III and Fluorol Yellow 088 to detect lipophilic substances [58]. Other leaves were preserved for 48 h in a FineFIX working solution (Milestone s.r.l., Bergamo, Italy) [59], dehydrated and paraffin-embedded. Eight-micron-thick cross sections were obtained using an automatic advanced rotative microtome (Leica RM 2255, Leica Biosystems, Heidelberg, Germany). After deparaffinization and rehydration, sections were stained with Hematoxylin–Eosin and with Toluidine Blue pH 4.0 as metachromatic staining [60,61] to carry out anatomical and histological characterization. Observations were made with a Leica DM 2000 fluorescence microscope equipped with an H3 filter (excitation filter BP 420–490 nm) (Leica Microsystems, Wetzlar, Germany) and with a ToupCam Digital Camera, CMOS Sensor 3.1 MP resolution (ToupTek Photonics, Hangzhou, China).

Small samples were also analysed by Scanning Electron Microscopy (SEM) to highlight micro-morphological and anatomical features, achieving a more detailed characterization. Fixed leaves were dehydrated in a graded ethanol series (70, 80, 90, and 100%) for 1 h each and subsequently critical point-dried using liquid carbon dioxide (CO_2_) (K850CPD 2M, Strumenti S.r.l., Roma, Italy). The dried specimens were then sectioned and mounted on aluminium stubs using two-sided adhesive carbon tape and covered with a 10 nm layer of gold particles. The examination was performed under a VEGA3-Tescan-type LMU microscope (Tescan USA Inc., Cranberry Twp, PA, USA), operating at an accelerating voltage of 20 kV.

### 4.3. Extraction of Essential Oils

Branches and leaves were reduced to fragments and then subjected to hydro-distillation for 3 h, as reported in the European Pharmacopoeia [62]. The EOs were dissolved in *n*-hexane, dried over anhydrous sodium sulphate, and stored under N_2_ at 4 °C in the dark until analysis.

### 4.4. Analysis of Essential Oils

Analytical gas chromatography was conducted on a Perkin–Elmer Sigma-115 gas chromatograph accessorized with an FID and a data handling processor. The separation was obtained with an HP-5MS fused-silica capillary column (30 m × 0.25 mm i.d., 0.25 μm film thickness). The column temperature was 40 °C, with a 5 min initial hold, and then raised to 270 °C at 2 °C/min, 270 °C (20 min); splitless injection (1 μL of a 1:1000 *n*-hexane solution). The injector and detector temperatures were 250 and 290 °C, respectively. The analysis was also run by using a fused silica HP Innowax polyethylenglycol capillary column (50 m × 0.20 mm i.d., 0.25 μm film thickness). In both cases, He was employed as carrier gas (1.0 mL/min). GC–MS analyses were conducted with a Hewlett–Packard 5890 A gas chromatograph linked online to an HP mass selective detector (MSD 5970HP), equipped with a DB-5 fused-silica column (25 m × 0.25 mm i.d.; 0.33 μm film thickness). The ionization energy voltage was 70 eV; the electron multiplier energy was 2000 V. The gas-chromatographic conditions were those described above; transfer line 295 °C. Most of the components were identified by comparing their Kovats indices (Ki) with those of the literature [63,64,65,66] and by a careful analysis of the mass spectra compared to those of pure compounds available in our laboratory or to those present in the NIST 02 and Wiley 257 mass libraries [67]. The Kovats indices were determined in relation to a homologous series of *n*-alkanes (C10-C35), under the same operating conditions. For some compounds, the identification was confirmed by co-injection with standard samples. The components’ relative concentrations were calculated by peak area normalization. Response factors were not considered.

### 4.5. Phytotoxic Activity

The phytotoxic activity was evaluated on seed germination and radicle emergence/elongation of several plant species, including weeds (*Lolium multiflorum* Lam. and *Sinapis arvensis* L.) and horticultural crops (*Raphanus sativus* L. and *Cucumis sativus* L.), selected for their easy and well-known germinability. *R. sativus* and *C. sativus* seeds were purchased from Blumen group s.r.l. (Milano, Italy); *L. multiflorum* seeds were purchased from Fratelli Ingegnoli Spa (Milano, Italy); and the seeds of *S. arvensis* were collected from a wild population growing near the University campus in Fisciano (Salerno, Italy). The seeds were surface-sterilized in 95% ethanol for 15 s and sown in Petri dishes (Ø = 90 mm) on three layers of Whatman filter paper. They were impregnated with (1) 7 mL of deionized water; (2) 7 mL of water–acetone mixture (99.5:0.5, *v*/*v*), used as a negative control since the EO was dissolved in this mixture due to its lipophilicity; and (3) 7 mL of the EO solutions at different concentrations (1000, 500, 250, and 125 μg/mL). These concentrations were selected based on previous studies carried out in our laboratory [68,69]. Controls carried out with the water–acetone mixture alone (negative control) showed no differences in comparison to the controls in water alone. The germination conditions were 20 ± 1 °C, with a natural photoperiod. Seed germination was checked in Petri dishes every 24 h. A seed was considered germinated when the protrusion of the root became evident [70]. On the fifth day (after 120 h) for *R. sativus*, and on the tenth day (after 240 h) for the other seeds, the effects on germination and radicle elongation were determined. The radical length was measured in cm. Each evaluation was replicated three times using Petri dishes containing 10 seeds each. Data were expressed as the mean ± standard deviation for both germination and radicle elongation.

### 4.6. Plant Pathogens

The EOs were tested against two bacterial microorganisms including *Xanthomonas campestris* pv. *campestris (Xcc)*, the causal agent of black root disease that causes significant harvest losses in the Brassicaceae, and *Pseudomonas syringae* pv. *tomato* (*P. syringae*), the causal agent of bacterial speck of tomato.

### 4.7. Antimicrobial Activity

A double agar assay was developed for the antimicrobial assessment of the EO on the tested pathogens. Briefly, stock solutions of the EO were dissolved in ethanol (ratio EO:EtOH of 1:9); then, serial dilutions were prepared (100%, 10%, 0.1%, and 0.01% *v*/*v*) in distilled water. Next, 500 μL of *Xcc* and *P. syringae* strains, grown in Nutrient Broth (NB) (Sigma Aldrich, Milan, Italy) to the exponential phase, were added individually to tubes containing 4 mL of 0.7% agar (Sigma Aldrich, Milan, Italy). The bacterial suspension was poured onto the surface of a Petri dish containing nutrient agar and let to solidify. The bacterial agar plates were divided into four sections and 10 μL of each EO dilution and the solvent control were inoculated to three points in each quadrat, then incubated at 25 °C for 48 h for *Xcc* and 28° C for 24 h for *P. syringae* [71]. The experiment was performed in triplicate for each bacterial pathogen. To determine the minimum inhibitory concentration (MIC) of the EO, the following different concentrations were tested on the growth inhibition of the bacteria: undiluted [100% *v*/*v*] and diluted at 1:10, 1:100, and 1:1000. Moreover, 1 mL of each dilutant and 0.5 mL of the bacterial cultures in an exponential growth phase (10^8^ CFU/mL) were added to 8 mL of Nutrient Broth with a final volume of 9.5 mL and incubated under shaking conditions at 25 °C for 48 h for *Xcc* and 28 °C for 24 h for *P. syringae*. After incubation, bacterial growth was evaluated by spectrophotometric readings of the culture concentrations via a Model 680 Microplate Reader at 600 nm (Bio-Rad Laboratories, Segrate, Italy).

### 4.8. Antibiofilm Activity

The assessment of biofilm formation with different dilutions of the EO was evaluated using a crystal violet staining assay as previously reported [72]. Two hundred µL aliquots of *Xcc* in the exponential growth phase were inoculated into a 96-well polystyrene plate (ThermoFisher, Waltham, MA USA) then incubated for 72 h at 24 °C, in a static condition. After, each dilution of the EO was added at the same time. After 4 h, each well was rinsed with distilled water several times. To evaluate the biofilm amount, each well was treated with 0.1% crystal violet, incubated for 10 min at RT, and rinsed with distilled water. Subsequently, the crystal violet was solubilized with 20% (*v*/*v*) acetone and 80% (*v*/*v*) ethanol, then the samples were measured at 655 nm to evaluate the biofilm biomass using a Tecan Infinite 200 Pro microplate reader (Tecan, Männedorf, Switzerland). Every data point consisted of six replicates, performed in three autonomous tests.

### 4.9. Bioassay on Tomato Leaves Infected with Pseudomonas syringae pv. tomato

A set of thirty tomato plants, in the fourth true leaf stage, were sprayed with distilled water (negative control), *Artemisia* EO (10% and 0.1% *v*/*v*), EtOH solvent control, and a commercial copper-based phytosanitary product (CUPRAVIT BLU 35WG; recommended field dose 1.4 kg/ha). After 3 days, the same plants were spray-inoculated with a suspension of *P. syringae* (10^6^ CFU/mL water) in a volume of ca. 30 mL per plant. Treated plants were incubated in a greenhouse with high humidity to favour stomatal penetration by the pathogen. For each leaflet, a disease severity index (T = 3) was calculated using the McKinney index [73] and multiplied by 100 to convert the 0–1 range to a percentage of disease.

### 4.10. Statistical Analysis

Statistical analysis of the activity in vitro was performed by one-way analysis of variance (ANOVA) using GraphPad Prism 6.0 (Software Inc., San Diego, CA, USA), expressed as mean ± standard deviation (S.D.). The results were compared to the untreated control and considered statistically significant, by Dunnett’s test, when *p* < 0.05 (* *p* < 0.05, ** *p* < 0.01, *** *p* < 0.001, **** *p* < 0.0001). The non-transformed values of the McKinney indexes were analysed by a one-way analysis of variance (ANOVA), and the significance of the differences was calculated by Tukey’s test (* *p* < 0.05, ** *p* < 0.01, *** *p* < 0.001, **** *p* < 0.0001).

## 5. Conclusions

Two kinds of secretory structures, namely, the glandular trichomes on the leaves and the secretory ducts in young stems, petioles, and leaves were observed in *A. arborescens*. The hydro-distillation of these plant portions gave an EO, rich in oxygenate monoterpenes such as *trans*-thujone and camphor, showing variable phytotoxicity on weeds or crops. In particular, a remarkable inhibitory activity on the two weed species tested was observed. Therefore, this EO represents a good candidate for further studies aimed at evaluating its activity in post-emergence and in vivo plant assays. The EO also showed antimicrobial activity both in vitro and *in planta*. Interestingly, inhibition was observed in vitro on growth of two plant pathogenic bacteria responsible for significant crop losses including *Xanthomonas campestris* pv. *campestris* and *Pseudomonas syringae* pv. *tomato*. It also produced a consistent reduction in biofilm biomass and a decrease in disease symptoms after *in planta* treatment. Overall, our data indicate that the EO of *A. arborescens* can be a promising treatment against pathogenic bacteria, reducing the symptoms of the disease in a manner comparable to commercial pesticides with minimal phytotoxicity in host plants. Nowadays, the search for new natural products that can replace synthetic pesticides to safeguard the environment and the health of consumers is becoming increasingly popular. Therefore, the results from this study open interesting perspectives for the use of *A. arborescens* EO as a potential alternative biocontrol product in organic agriculture.

## Figures and Tables

**Figure 1 plants-13-00825-f001:**
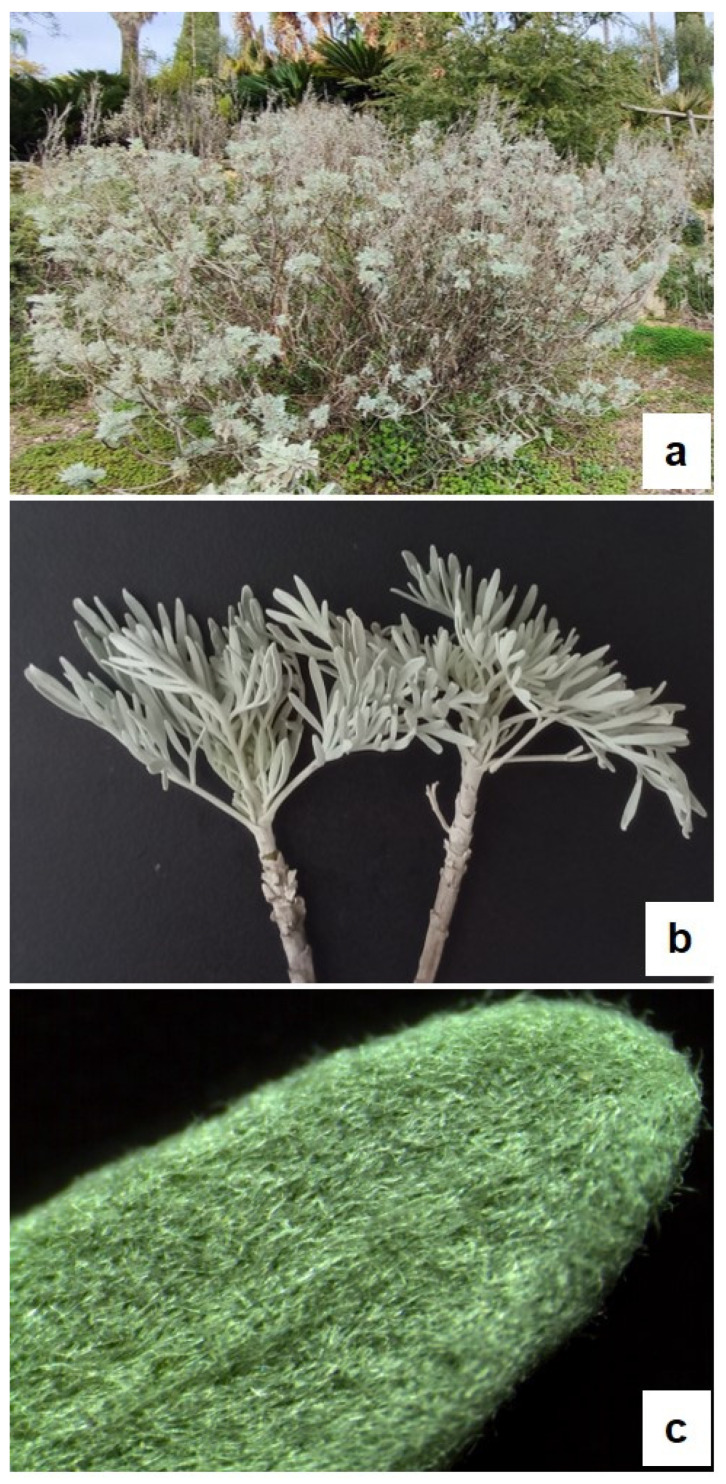
*Artemisia arborescens* (Vaill.) L. (**a**) Plant growing at the Hanbury Botanical Gardens (Ventimiglia, Italy); (**b**) small branches with petiolate and deeply divided leaves; and (**c**) detail of a leaf showing a dense cover of trichomes.

**Figure 2 plants-13-00825-f002:**
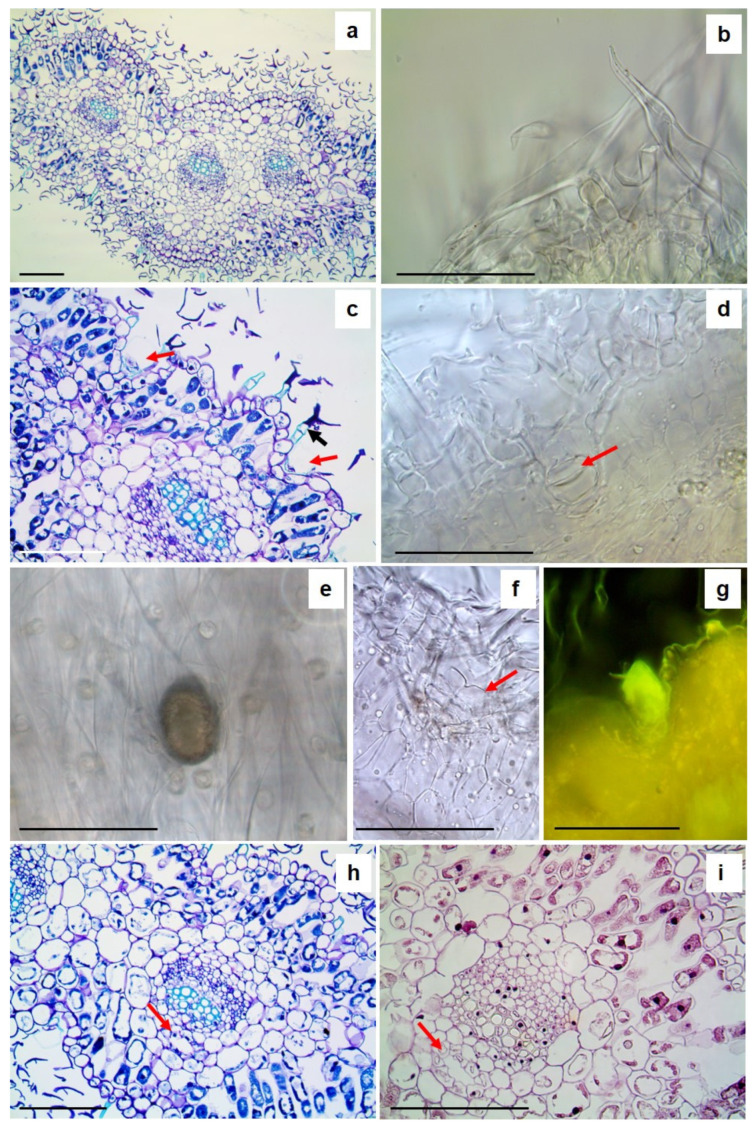
LM micrographs of a leaf of *A. arborescens*: (**a**,**c**,**h**,**i**) semithin sections and (**b**,**d**,**e**–**g**) handmade sections. (**a**) A transversal section stained with TBO showing the general anatomy of the leaf; (**b**) detail of T-shaped non-glandular trichomes; (**c**) a transversal section stained with TBO highlighting capitate glandular trichomes deeply embedded into the leaf surface (red arrows) and T-shaped non-glandular trichomes with suberin- or cutin-like hydrophobic substances on the side walls of the neck cells (black arrow); (**d**) a capitate glandular trichome in transversal section cleared with Chloral Hydrate solution (red arrow); (**e**) a capitate glandular trichome filled with essential oil on the leaf epidermis; (**f**) a biseriate capitate glandular trichome in transversal section; (**g**) a glandular trichome stained by Fluoral Yellow, revealing the presence of lipophilic substances; and (**h**,**i**) secretory ducts near the xilematic portion of the vascular bundles (red arrow). Bars 100 µm.

**Figure 3 plants-13-00825-f003:**
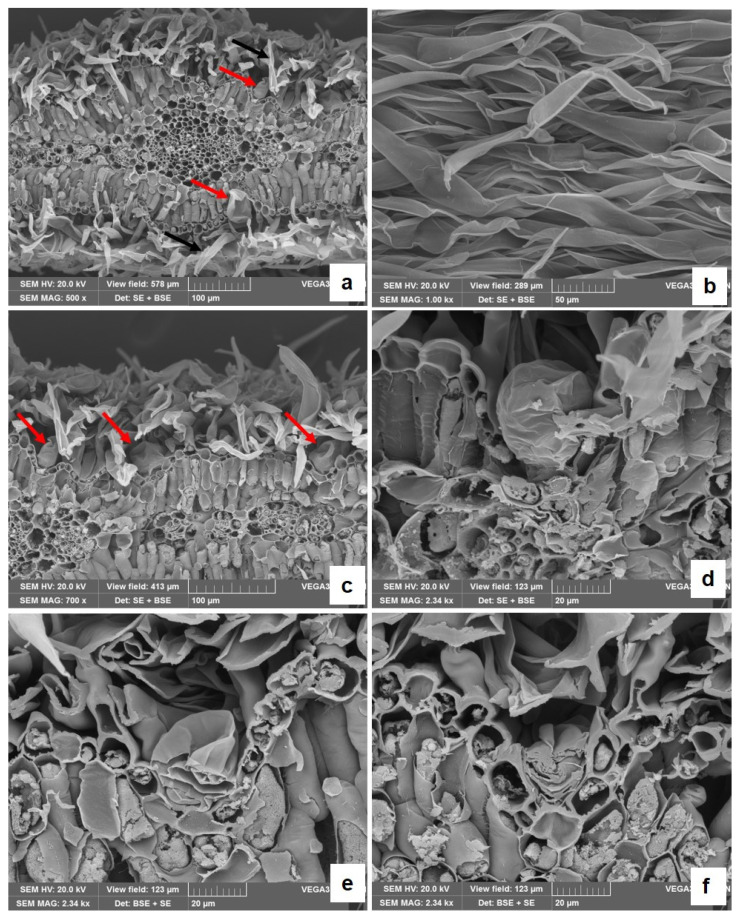
SEM micrographs of a leaf of *A. arborescens.* (**a**) A transversal section of the leaf showing the presence of many T-shaped non-glandular trichomes (black arrows) and of glandular trichomes (red arrows) on both leaf surfaces; (**b**) a higher magnification of T-shaped non-glandular trichomes on the leaf epidermis; (**c**) a leaf transversal section showing capitate glandular trichomes deeply embedded into the leaf surface (red arrows) and partially hidden by the dense covering of the non-glandular trichomes; and (**d**–**f**) capitate glandular trichomes at higher magnification. In (**f**), the section shows the secretory cells within the glandular head.

**Figure 4 plants-13-00825-f004:**
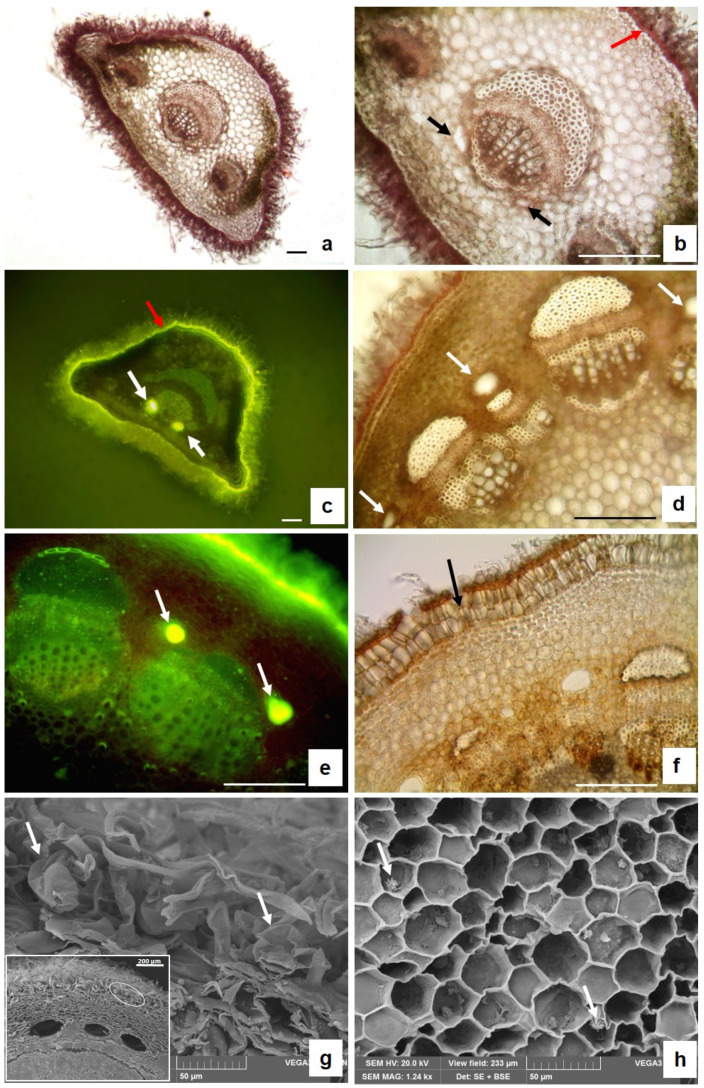
LM (**a**–**f**) and SEM (**g**,**h**) micrographs of transversal sections of a leaf petiole (**a**–**c**) and young stem (**d**,**h**). (**a**) General petiole anatomy; (**b**) detail of the petiole showing two secretory ducts on both sides of the main vascular bundle (black arrows) and a thick cuticle over the epidermis, stained red with Sudan III (red arrows); (**c**) brilliant yellow fluorescence of the lipophilic substances of the cuticle (red arrows) and of the essential oil inside the secretory ducts stained with FY (white arrows); (**d**) young stem anatomy revealing the presence of secretory ducts in the cortex near the vascular bundle (white arrows); (**e**) brilliant yellow fluorescence of the essential oil inside secretory ducts stained with FY (arrows); (**f**) periderm (black arrow) in the older stem; (**g**) glandular trichomes on the stem surfaces (white arrows); the insert shows the location of the glandular trichomes; and (**h**) small crystal druses inside the pith parenchyma cells (white arrows).

**Figure 5 plants-13-00825-f005:**
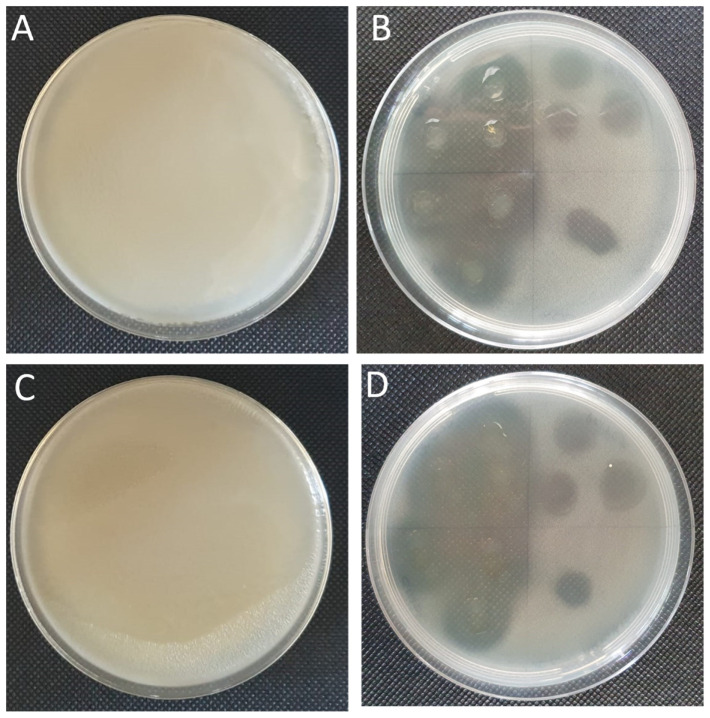
Inhibition of bacterial colony growth by *Artemisia* EO as noted by the inhibition zone formation on *Xcc* and *P. syringae* plaques in NB soft agar. (**A**) *Xcc* untreated, (**B**) *Xcc* with EO, (**C**) *P. syringae* untreated, and (**D**) *P. syringae* with EO. The Petri dish (**B**,**D**) was divided into 4 sections, and three point-inoculations were performed with each of the different EO serial dilutions (100%, 10%, 0.1%, and 0.01% *v*/*v*). The dilutions were inoculated respectively to each quadrant in the same order, starting in the lower left section and rotating in a clockwise direction.

**Figure 6 plants-13-00825-f006:**
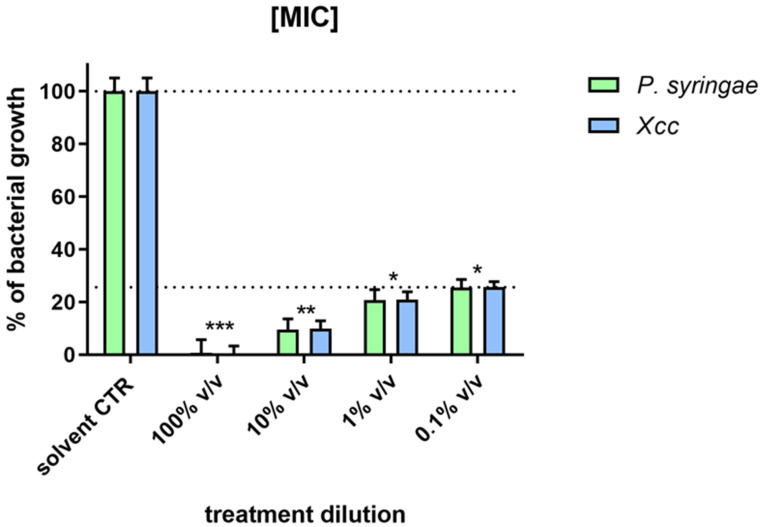
Antibacterial activity, the minimum inhibitory concentration [MIC] of different EO concentrations (undiluted and diluted) on culture growth of *P. syringae* (24 h) and *Xcc* (48 h), after treatment. Statistical analysis was performed with the absorbance compared to the untreated control and considered statistically significant when *p* < 0.05 (* *p* < 0.05, ** *p* < 0.01, *** *p* < 0.001) according to one-way ANOVA and multiple comparisons.

**Figure 7 plants-13-00825-f007:**
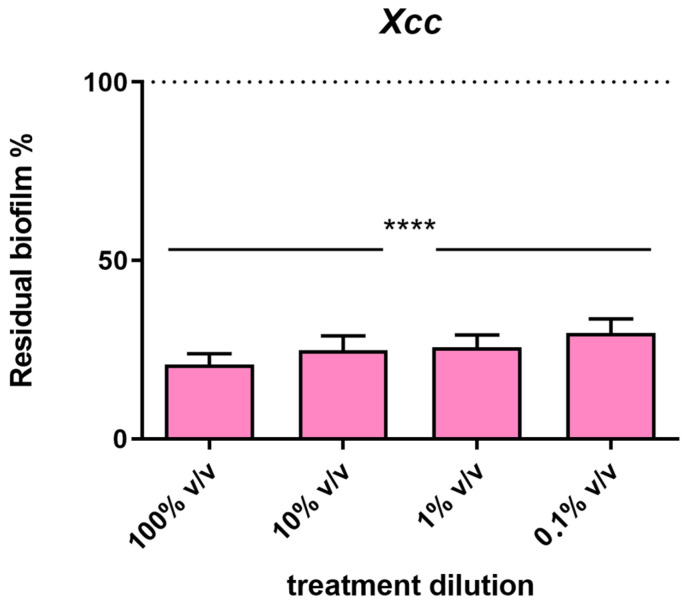
Activity of the EO on biofilm formation of *Xcc* after 72 h of incubation. Determination by spectrophotometer readings at 655 nm of the crystal violet intensity associated with the biofilm produced by the bacteria. Data are expressed as a percentage of residual biofilm. Each value indicates the mean ± SD of three independent experiments. Statistical analysis was performed with the absorbance compared to the untreated control and considered statistically significant at *p* < 0.05 (**** *p* < 0.0001) according to one-way ANOVA multiple comparisons.

**Figure 8 plants-13-00825-f008:**
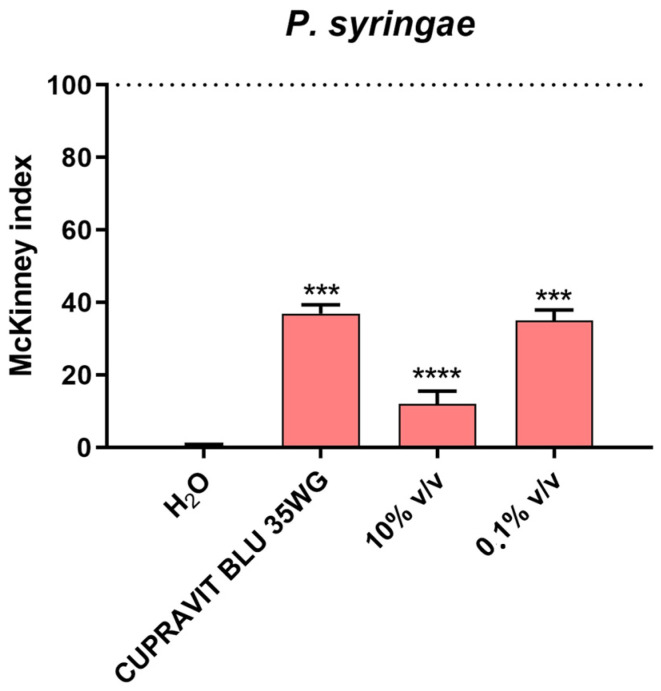
Effect of EO treatments on disease severity caused by *P. syringae* inoculated onto the leaves of tomato plants. Disease severity was measured by the McKinney index. *P. syringae* was inoculated at 10^6^ CFU/mL, EO at 10%, and 0.1% *v*/*v* and CUPRAVIT BLUE 35WG at 1.4 kg/ha. Values are the mean ± SD of three replicates (30 plants each) per treatment. Statistical analysis was performed with the absorbance compared to the untreated control, and data were considered statistically significant when *p* < 0.05 (*** *p* < 0.001, **** *p* < 0.0001) according to one-way ANOVA multiple comparisons.

**Table 1 plants-13-00825-t001:** Composition of the EO of *A. arborescens*.

	KI ^a^	KI ^b^	%	Identification ^c^
Tricyclene	844	1047	0.3	1, 2
α-Thujene	850	1020	0.1	1, 2
α-Pinene	855	1036	2.3	1, 2, 3
Camphene	868	1075	6.0	1, 2, 3
Butanoic acid, 2-methylpropyl ester	870	1174	0.1	1, 2
Sabinene	891	1115	2.4	1, 2
β-Pinene	910	1120	0.8	1, 2, 3
α-Phellandrene	918	1177	1.8	1, 2, 3
α-Terpinene	929	1170	0.6	1, 2, 3
Propanoic acid, 2-methyl-, 3-methylbutyl ester	934		0.2	1, 2
*p*-Cymene	938	1250	2.4	1, 2, 3
Eucalyptol	942	1210	2.2	1, 2, 3
γ-Terpinene	970	1221	1.0	1, 2, 3
Terpinolene	997	1267	0.2	1, 2, 3
*cis*-Thujone	1009	1430	1.2	1, 2
*trans*-Thujone	1020	1442	24.2	1, 2
α-Campholenal	1032	1485	0.2	1, 2
*allo*-Ocimene	1041	1388	0.1	1, 2
Camphor	1050	1491	18.9	1, 2, 3
Pinocarvone	1063	1586	0.1	1, 2, 3
Borneol	1067	1715	0.2	1, 2, 3
Terpinen-4-ol	1079	1590	0.7	1, 2, 3
Isocitral	1084	1690	0.1	1, 2
α-Terpineol	1093	1661	0.3	1, 2
*cis*-Chrysantenyl acetate	1098		0.2	1, 2
4-Decen-1-ol	1133		0.2	1, 2
Thymol	1199	2172	0.6	1, 2, 3
δ-Elemene	1219	1479	0.1	1, 2
α-Copaene	1255	1477	0.4	1, 2
Isobornyl propanoate	1258		0.1	1, 2
β-Bourbonene	1263	1498	0.3	1, 2
β-Elemene	1266	1579	0.1	1, 2
β-Gurjunene	1273	1615	1.2	1, 2
Aromadendrene	1298	1631	6.5	1, 2
*cis*-Muurola-3,5-diene	1299		0.7	1, 2
α-Humulene	1322	1641	0.8	1, 2
γ-Gurjunene	1346		0.7	1, 2
Germacrene D	1350	1712	1.5	1, 2
*cis*-β-Guaiene	1354	1651	0.2	1, 2
*trans*-Muurola-4(14),5-diene	1365		0.3	1, 2
γ-Amorphene	1383		0.5	1, 2
δ-Amorphene	1393	1751	0.5	1, 2
α-Cadinene	1400	1753	0.1	1, 2
Caryophyllene oxide	1443	2000	3.0	1, 2
Aristolene epoxide	1453		0.2	1, 2
Caryophylla-4(12),8(13)-dien-5α-ol	1498	2324	0.8	1, 2
α-Cadinol	1507	2256	1.3	1, 2
Guaia-3,10(14)-dien-11-ol	1538		0.1	1, 2
*cis*-Z-α-Bisabolene epoxide	1550		0.1	1, 2
Cedren-13-ol, 8-	1837	2359	5.2	1, 2
*trans*-Nuciferol	1839		4.2	1, 2
**Total**			**96.3**	
Monoterpene hydrocarbons			18.0	
Oxygenated monoterpenes			49.0	
Sesquiterpene hydrocarbons			13.9	
Oxygenated sesquiterpenes			14.9	
Others			0.5	

^a,b^: Kovats retention indices determined relative to a series of *n*-alkanes (C10–C35) on the apolar HP-5 MS and the polar HP Innowax capillary columns, respectively; ^c^: identification method: 1 = comparison of the Kovats retention indices with published data, 2 = comparison of mass spectra with those listed in the NIST 02 and Wiley 275 libraries and with published data, and 3 = coinjection with authentic compounds.

**Table 2 plants-13-00825-t002:** Phytotoxic activity of *A. arborescens* EO on seed germination and root growth.

Number of Germinated Seeds (A) and Percent Inhibition (B)								
µg/mL	*L. multiflorum*		*S. arvensis*		*R. sativus*		*C. sativus*	
	A	B	A	B	A	B	A	B
Control	10.0 ± 0.0	0	9.7 ± 0.6	0	10.0 ± 2.0	0	8.7 ± 0.6	0
125	7.0 ± 1.0 *	30.0	3.3 ± 0.6 ****	65.9	3.0 ± 1.0 *	70.0	9.7 ± 0.6	−111.5
250	6.3 ± 1.5 **	37.0	2.0 ± 1.0 ****	70.3	1.7 ± 1.2 **	83.0	9.3 ± 1.2	−106.0
500	1.7 ± 1.5 ****	83.0	10.0 ± 0.0	−103.1	0.0 ± 0.0 ***	100.0	7.0 ± 2.0	19.5
1000	0.0 ± 0.0 ****	100.0	10.0 ± 0.0	−103.1	0.0 ± 0.0 ***	100.0	4.0 ± 1.0 **	54.0
**Radical Length (cm—A) and Percent Inhibition (B)**								
**µg/mL**	** *L. multiflorum* **		** *S. arvensis* **		** *R. sativus* **		** *C. sativus* **	
	**A**	**B**	**A**	**B**	**A**	**B**	**A**	**B**
Control	0.6 ± 0.4	0	0.8 ± 0.6	0	0.9 ± 0.3	0	1.0 ± 0.4	0
125	0.4 ± 0.2	33.3	0.2 ± 0.1	25.0	0.7 ± 0.0	22.2	2.3 ± 0.5 *	−230.0
250	0.3 ± 0.1	50.0	0.2 ± 0.0	25.0	0.0 ± 0.0 ****	100.0	1.5 ± 0.4	−150.0
500	0.0 ± 0.0 *	100.0	0.0 ± 0.0 *	100.0	0.0 ± 0.0 ****	100.0	1.6 ± 0.5	−160.0
1000	0.0 ± 0.0 *	100.0	0.0 ± 0.0 *	100.0	0.0 ± 0.0 ****	100.0	0.2 ± 0.1	80.0

The results are the mean of three experiments ± standard deviation. * *p* < 0.05; ** *p* < 0.01; *** *p* < 0.001; **** *p* < 0.00001 compared with the control (ANOVA followed by Dunnett’s multiple comparison test). Control = 0 µg/mL. Column B indicates a stimulation of germination or radical growth.

## Data Availability

The data presented in this study are available in this published paper.

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
