# Peer review of "Artemisia arborescens (Vaill.) L.: Micromorphology, Essential Oil Composition, and Its Potential as an Alternative Biocontrol Product"

_plants, 2024, doi:10.3390/plants13060825_

Round 1

Reviewer 1 Report

Comments and Suggestions for Authors

Dear Authors,

I attach a review of the article „ Artemisia arborescens (Vaill.) L.: micromorphology, essential oil composition and its phytotoxic and antimicrobic activities”.

General remarks

·        Section Introduction needs to be improved. The gaps in science and the need for the research presented in the article should be clearly presented.

·        In section Introduction research of alternative biocontrol product against weeds and plant pathogens is omitted. The need for research should be highlighted. Must be improved.

·        The next problem is secretory structures where the EO is produced and accumulated. Haven't there been such studies before? Must be improved.

·        In section Materials and Methods the description of statistics is inaccurate and enigmatic. Subsection Statistical Analyses should be given and statistic tools used in particular experiments should be clearly described.

·        Many of the comments below concern statistical analyses.

·        Standard is: *p < 0.05; ** p < 0.01; *** p < 0.001.

·        References must be improved.

Comments

Line 14: …Abstract Artemisia arborescens is a Mediterranean aromatic shrub…

Rev: shrub or herb? what life form?

Line 38: …This aromatic multi-branched shrub…

Rev: shrub or herb? what life form?

Line 145: …The EO oil yield…

Rev: “oil” repetition

Line 146: …other Artemisa species…

Rev: should be: Artemisia

Line 150-156: …monoterpenes trans-thujone (24.2%) and camphor (18.9%) resulted the main components of the EO. Among monoterpene hydrocarbons the main components were camphene (6.0%), p-cymene (2.4%), sabinene (2.4%), and α- pinene (2.3%). Sesquiterpene fraction was almost equally divided among hydrocarbons (14.9%) and oxygenated com pounds (13.9%). Aromadendrene (6.5%) was the main component among sesquiterpene hydrocarbons, trans-nuciferol (4.2%) and caryophyllene oxide (3.0%) among oxygenated sesquiterpenes (Table 1)…

Rev: spelling of EO names should be the same in all manuscript; see Table l

Line 180: …Table 2 Phytotoxic activity of A. arborescens EO…

Rev: data requires verification and a critical approach;

Line 181-182: … **** p < 0.00001…

Rev: ?

Rev: Table 2 Number of germinated seeds S. arvensis at 250 (μg/mL) germination is 20% and at 1000 (μg/mL) – 100%?

Line 200: Figure 6:

Rev: “treatment diluition” – should be corrected

Line 201: … (MIC)…

Rev: the same type of parenthesis should be on picture

Line 202: …P. syringae…

Rev: name on picture should be the same

Line 201-203: …Antibacterial activity, the minimum inhibitory concentration (MIC) of different EO concentrations (undiluted and diluted) on culture growth of P. syringae and Xcc, 24 and 48 h after treatment…

Rev: where is 24h and 48h?

Line 203-205: Statistical analysis was performed with the absorbance compared to the untreated control and considered statistically significant when p < 0.05 (* p < 0.05, ** p < 0.01, *** p <  0.001,) according to two-way ANOVA multiple comparisons.

Rev: “compared to the untreated control” - this statement refers to the comparison of 2 groups, while ANOVA is a test in which more than 2 groups are compared. Please correct and refer to the correct test for the 2 groups.

Line 223:..P. syringae…

Rev: italics

Line 223-228: …The results of the trial showed a statistically significant decrease in disease symptoms (84%) with treatment at 10% v/v of EO, but it seems to be phytotoxic for the plants. Interestingly, when the plants were treated with 0.1% v/v of EO the decrease in disease symptoms on  plants are comparable with that of the commercial pesticide (75%) (Figure 8), without phytotoxic symptoms.

Rev: in my opinion the commercial pesticide and 0.1% v/v are ca 65%

Rev: what about H2O?

Line 234-237: Statistical analysis was performed with the absorbance compared to the untreated control and data was considered statistically significant when p < 0.05 (*** p < 0.001, *** p < 0.0001) according to two-way ANOVA multiple comparisons.

Rev: “compared to the untreated control” - this statement refers to the comparison of 2 groups, while ANOVA is a test in which more than 2 groups are compared. Please correct and refer to the correct test for the 2 groups.

Rev: should be **** p < 0.0001

Line 265-266: …Some studies are available in the literature regarding the composition of the EO of A. arborescens, but only a few studies concern the EOs obtained from species growing in Italy…

Rev: The cited literature indicates that there were a lot of these studies

Line 425: …the causal agent of bacterial speck of tomato…

Rev: dot

Line 493: …potential alternative biocontrol product in organic agriculture..

Rev: dot

Line 479-481: …Two secretory structures present in A. arborescens, namely the glandular trichomes on the leaves and the secretory ducts within the stem and petiole, are the sites where the EO, rich in oxygenate monoterpenes such as trans-thujone and camphor, is accumulated…

Rev: this is a suggestion that trans-thujone and camphor were located in the trichomes. No such studies have been conducted. EO profile was studied after hydro-distillation of branches and leaves. Should be redraft.

Line 484: …Xanthomonas campestris pv. campestris (Xcc),

Rev: (Xcc) is unnecessary

Line 522: …38:128…

Line 526: Ascensão, L.; Pais, M.S.S. Glandular Trichomes of Artemisia campestris (ssp. Maritima): Ontogeny and Histochemistry of 526 the Secretory Product. Bot Gaz 1987, 148, 221-227. doi: 10.1086/337650

Line 531-533: Bewley, J.D. Seed Germination and Dormancy. Plant Cell 1997, 9, 1055–1066. doi: 10.1105/tpc.9.7.1055 531

10. Bilia, A.R.; Santomauro, F.; Sacco, C.; Bergonzi, M.C.; Donato, R. Essential Oil of Artemisia annua L.: An Extraordinary 532 Component with Numerous Antimicrobial Properties. Evid Based Complement Alternat Med, 2014, 59819. doi: 533 10.1155/2014/159819

Line 546: Pharm Biol2015

Line 547-548:Das, S.; Vörös-Horváth, B.; Bencsik, T.; Micalizzi, G.; Mondello, L.; Horváth, G.; KÅ‘szegi, T.; Széchenyi, A. Antimicrobial 547 Activity of Different Artemisia Essential Oil Formulations. Molecules 2020, 25, 2390. doi: 10.3390/molecules25102390

Line 553-554: Duke, S.O.; Paul, R.N. Development and Fine Structure of the Glandular Trichomes of Artemisia annua L. Int J Plant Sci 553 1993, 154, 107–118. Doi: 10.1086/297096

Line 560: pseudomonas aeruginosa

it takes a lot of work

Comments on the Quality of English Language

aa

Author Response

Dear Authors,

I attach a review of the article “Artemisia arborescens(Vaill.) L.: micromorphology, essential oil composition and its phytotoxic and antimicrobic activities”.

General remarks

1) Section Introduction needs to be improved. The gaps in science and the need for the research presented in the article should be clearly presented.

Dear Reviewer thank you for your useful suggestion, we have improved the introduction accordingly and now we have better clarify the lack of data about issue considered in our study.

2) In section Introduction research of alternative biocontrol product against weeds and plant pathogens is omitted. The need for research should be highlighted. Must be improved.

We added new data and references highlighting the importance of biological control. We also emphasized the potential use of essential oils as an alternative to traditional pesticides in the Introduction.

3) The next problem is secretory structures where the EO is produced and accumulated. Haven't there been such studies before? Must be improved.

Referring to A. arborescens secretory structures, the only study present in literature is the one of Janackovic et al 2019, who described these secretory tissues comparing them with those found in other Artemisia species. On the contrary, there is many references related to glandular trichomes and secretory ducts present in many other species belonging to Artemisia genus. Therefore, according to your suggestion we have added several citations about this issue.

We have also modified Fig. 2 by inserting a fluorescence microscope photograph showing that the secretion of glandular trichomes reacts to the Fluoral Yellow 088 staining like that of the secretory ducts, highlighting a similar composition.

4) In section Materials and Methods the description of statistics is inaccurate and enigmatic. Subsection Statistical Analyses should be given and statistic tools used in particular experiments should be clearly described.

We have added a paragraph in the Materials and Methods section that describes the statistics and specifications for each experiment.

5) Many of the comments below concern statistical analyses.

Standard is: *p < 0.05; ** p < 0.01; *** p < 0.001.

We have reviewed and adjusted all statistical information to ensure its accuracy and consistency.

6) References must be improved.

We have improved the references by adding some new references, both in the Iintroduction and in the Discussion.

Comments

7) Line 14: …Abstract Artemisia arborescens is a Mediterranean aromatic shrub…

Rev: shrub or herb? what life form?

8) Line 38: …This aromatic multi-branched shrub…

Rev: shrub or herb? what life form?

We have delayed the adjective “aromatic”, which is frequently associated to herbs, in order to clarify that A.arborescens is not an herb but an evergreen shrub characterized by a strong scent. (See floraveg.eu/en/taxon/overview/Artemisia%20arborescens).

9) Line 145: …The EO oil yield…

Rev: “oil” repetition

Done

10) Line 146: …other Artemisa species…

Rev: should be: Artemisia

We have modified.

11) Line 150-156: …monoterpenes trans-thujone (24.2%) and camphor (18.9%) resulted the main components of the EO. Among monoterpene hydrocarbons the main components were camphene (6.0%), p-cymene (2.4%), sabinene (2.4%), and α- pinene (2.3%). Sesquiterpene fraction was almost equally divided among hydrocarbons (14.9%) and oxygenated com pounds (13.9%). Aromadendrene (6.5%) was the main component among sesquiterpene hydrocarbons, trans-nuciferol (4.2%) and caryophyllene oxide (3.0%) among oxygenated sesquiterpenes (Table 1)…

Rev: spelling of EO names should be the same in all manuscript; see Table l

Dear Reviewer, in accordance with the rules of the journal, we have followed the standard compilation of the Table with the names of chemical compounds in capital letters. Also in other works published on Plants the names of chemical compounds in the Tables are cited with a capital letter, while in the text they are written with a lowercase letter. We defer to the opinion of the editorial committee.

12) Line 180: …Table 2 Phytotoxic activity of A. arborescens EO…

Rev: data requires verification and a critical approach.

This phenomenon is due to hormesis and as explained further below see answer 14

13) Line 181-182: … **** p < 0.00001…

Rev: ?

Sorry, we made a typo. We have now changed to the correct one.

14) Rev: Table 2 Number of germinated seeds S. arvensis at 250 (μg/mL) germination is 20% and at 1000 (μg/mL) – 100%?

Thanks for the suggestion; this phenomenon is due to hormesis, a two-step biological response in which a low dose of a biological agent shows inhibitory activity and a high dose the opposite effect.

We have included a sentence in the Discussions to explain it.

15) Line 200: Figure 6:

Rev: “treatment diluition” – should be corrected

We have modified.

16) Line 201: … (MIC)…

Rev: the same type of parenthesis should be on picture

Done, we have modified the type of parenthesis in the caption of the figure.

17) Line 202: …P. syringae…

Rev: name on picture should be the same

Thank you for your suggestion, we have modified the name in the figure.

18) Line 201-203: …Antibacterial activity, the minimum inhibitory concentration (MIC) of different EO concentrations (undiluted and diluted) on culture growth of P. syringae and Xcc, 24 and 48 h after treatment…

Rev: where is 24h and 48h?

The growth time of each bacterium on the plate is represented by 24h and 48h. We have revised the figure legend for clarity.

19) Line 203-205: Statistical analysis was performed with the absorbance compared to the untreated control and considered statistically significant when p < 0.05 (* p < 0.05, ** p < 0.01, *** p <  0.001,) according to two-way ANOVA multiple comparisons.

Rev: “compared to the untreated control” - this statement refers to the comparison of 2 groups, while ANOVA is a test in which more than 2 groups are compared. Please correct and refer to the correct test for the 2 groups.

We have modified the statistical information, accordingly.

20) Line 223:..P. syringae…

Rev: italics

Done.

21) Line 223-228: …The results of the trial showed a statistically significant decrease in disease symptoms (84%) with treatment at 10% v/v of EO, but it seems to be phytotoxic for the plants. Interestingly, when the plants were treated with 0.1% v/v of EO the decrease in disease symptoms on plants are comparable with that of the commercial pesticide (75%) (Figure 8), without phytotoxic symptoms.

Rev: in my opinion the commercial pesticide and 0.1% v/v are ca 65%

Rev: what about H2O?

We incorporated your suggestion to modify the disease percentage and added information about the plant group treated with H2O.

22) Line 234-237: Statistical analysis was performed with the absorbance compared to the untreated control and data was considered statistically significant when p < 0.05 (*** p < 0.001, *** p < 0.0001) according to two-way ANOVA multiple comparisons.

Rev: “compared to the untreated control” - this statement refers to the comparison of 2 groups, while ANOVA is a test in which more than 2 groups are compared. Please correct and refer to the correct test for the 2 groups.

Rev: should be **** p < 0.0001

We have modified the statistical information.

23) Line 265-266: …Some studies are available in the literature regarding the composition of the EO of A. arborescens, but only a few studies concern the EOs obtained from species growing in Italy…

Rev: The cited literature indicates that there were a lot of these studies

The sentence was misspelled, thanks for pointing it out. Now the sentence has been modified as follows:

”Some studies are available in the literature regarding the composition of the EO of A. arborescens, and some of these studies concern the EOs obtained from species growing in Italy”.

24) Line 425: …the causal agent of bacterial speck of tomato…

Rev: dot

We have modified.

25) Line 493: …potential alternative biocontrol product in organic agriculture..

Rev: dot

We have modified.

26) Line 479-481: …Two secretory structures present in A. arborescens, namely the glandular trichomes on the leaves and the secretory ducts within the stem and petiole, are the sites where the EO, rich in oxygenate monoterpenes such as trans-thujone and camphor, is accumulated…

Rev: this is a suggestion that trans-thujone and camphor were located in the trichomes. No such studies have been conducted. EO profile was studied after hydro-distillation of branches and leaves. Should be redraft.

Thank you for this suggestion, the sentence was misspelled. We have redraft it and we have improved the Conclusions.

27) Line 484: …Xanthomonas campestris pv. campestris (Xcc),

Rev: (Xcc) is unnecessary

Thanks for your suggestion. We have removed it.

References

28) Line 522: …38:128…

Line 526: Ascensão, L.; Pais, M.S.S. Glandular Trichomes of Artemisia campestris (ssp. Maritima): Ontogeny and Histochemistry of 526 the Secretory Product. Bot Gaz 1987,148, 221-227. doi: 10.1086/337650

29) Line 531-533: Bewley, J.D. Seed Germination and Dormancy. Plant Cell 1997, 9, 1055–1066. doi: 10.1105/tpc.9.7.1055 531

30) 10. Bilia, A.R.; Santomauro, F.; Sacco, C.; Bergonzi, M.C.; Donato, R. Essential Oil of Artemisia annua L.: An Extraordinary 532 Component with Numerous Antimicrobial Properties. Evid Based Complement Alternat Med, 2014, 59819. doi: 533 10.1155/2014/159819

31) Line 546: Pharm Biol2015

32) Line 547-548:Das, S.; Vörös-Horváth, B.; Bencsik, T.; Micalizzi, G.; Mondello, L.; Horváth, G.; KÅ‘szegi, T.; Széchenyi, A. Antimicrobial 547 Activity of Different Artemisia Essential Oil Formulations. Molecules 2020, 25, 2390. doi: 10.3390/molecules25102390

33) Line 553-554:Duke, S.O.; Paul, R.N. Development and Fine Structure of the Glandular Trichomes of Artemisia annua L. Int J Plant Sci 553 1993, 154, 107–118. Doi: 10.1086/297096

34) Line 560: pseudomonas aeruginosa

it takes a lot of work

The reference list have been reviewed according to your suggestion and to the Journal's guidelines.

Reviewer 2 Report

Comments and Suggestions for Authors

Dear authors,

Below, you'll find some remarks and recommendations to improve your manuscript:

Title: Please completely revise the title to make it more attractive. you can use "botanical pesticide" or "potential alternative biocontrol product" . Replace "antimicrobic" .

Abstract: The abstract is too lengthy and repeats the results section. Clearly state the aim of your work and the issues addressed. Explain why you pursued the development of a biocontrol product.

Introduction: The introduction is too brief. Include the issues surrounding your subject matter, such as weed control, plant pathogens, or the use of chemical pesticides. Detail what this work aims to achieve and how. 

Results and Discussion:

Elaborate on the connection between the micromorphology of Artemisia and its biological activities.

Convert phytotoxic activity results into percentage of inhibition.

For antimicrobial activity, in Figure 5, include a picture of non-treated plant pathogens to highlight the effects better.

Materials and Methods:

Reformulate line 407  "verifying the germinability of the seeds."

Explain the choice of concentration of EO.

Conclusion: Include the results of the phytotoxic effect. Address the possibility of post-emergence testing and offer future perspectives for your work.

Best regards, 

Author Response

Answers to the 2°Review

Dear authors,

Below, you'll find some remarks and recommendations to improve your manuscript:

Title:

1) Please completely revise the title to make it more attractive. you can use "botanical pesticide" or "potential alternative biocontrol product" . Replace "antimicrobic" .

Thank you for this important suggestion, we have modified the Title accordingly and now it seems to be more attractive.

Abstract:

2) The abstract is too lengthy and repeats the results section. Clearly state the aim of your work and the issues addressed. Explain why you pursued the development of a biocontrol product.

We have modified the Abstract by reducing the exposition of the results. We have also better clarified the aim of the work and the importance of looking for new biocontrol products.

Introduction:

3) The introduction is too brief. include the issues surrounding your subject matter, such as weed control, plant pathogens, or the use of chemical pesticides. Detail what this work aims to achieve and how.

In the Introduction we have added new information and references relating to the importance of biological control and the potential use of essential oils as an alternative to traditional pesticides.

Furthermore, details about the purpose and importance of the present study have been added

Results and Discussion:

4) Elaborate on the connection between the micromorphology of Artemisia and its biological activities.

Our study is based on a pharmacognostic approach, including both anatomical and micro-morphological study, as well as phytochemical analysis. Identifying the type and location of secretory structures in a plant species is important at a taxonomic level as well as for the control of the raw material before the subsequent extraction/distillation phases. In the Discussion we have now better clarified that the secreting system present both inside the organs (ducts) and on the organs surface (glandular trichomes) is involved in the production of a secretion rich of several bioactive compounds. Among them essential oil play an important role for plant adaptation to the specific environment, which include the allelopathic effects and the plant defence against pathogens.

5) Convert phytotoxic activity results into percentage of inhibition.

In table 2 we added the percent inhibition (column B) to make the effect of the essential oil on the seeds more evident.

6) For antimicrobial activity, in Figure 5, include a picture of non-treated plant pathogens to highlight the effects better.

We have followed your suggestion and included the images of the control plates in the figure.

Materials and Methods:

7) Reformulate line 407  "verifying the germinability of the seeds."

Done, the sentence has been deleted.

8) Explain the choice of concentration of EO.

We supported the choice of the different concentrations used by referring to previous works in which similar concentrations of various EOs were tested on both weeds and crops as well as on pathogenic bacteria.

Conclusion:

9) Include the results of the phytotoxic effect. Address the possibility of post-emergence testing and offer future perspectives for your work.

Done, we changed the sentence as follows: “In particular, a remarkable inhibitory activity on the two weeds tested was observed. Therefore, this EO represents a good candidate for further studies aimed at evaluating its activity in post-emergence and in vivo assays”.

Round 2

Reviewer 1 Report

Comments and Suggestions for Authors

a